

# 3D Water Vapor Field in the Atmospheric Boundary Layer Observed with Scanning Differential Absorption Lidar

F. Späth, A. Behrendt, S. K. Muppa, S. Metzendorf, A. Riede, and V. Wulfmeyer

University of Hohenheim, Institute of Physics and Meteorology, Garbenstr. 30, 70599 Stuttgart, Germany

5   Correspondence to: F. Späth (f.spaeth@uni-hohenheim.de)

**Abstract.** The scanning differential absorption lidar (DIAL) of the University of Hohenheim (UHOH) determines fields of the atmospheric water vapor number density with a temporal resolution of a few seconds and spatial resolution of up to a few tens of meters. We present three case studies which show that this high resolution combined with 2- and 3-dimensional scans allows for new insights in the 3-dimensional structure of the water vapor field in the atmospheric boundary layer (ABL). In spring 2013, the UHOH DIAL was operated within the scope of the HD(CP)$^2$ Observational Prototype Experiment (HOPE) in western Germany. HOPE was part of the project High Definition of Clouds and Precipitation for advancing Climate Prediction (HD(CP)$^2$). Range-height indicator (RHI) scans of the UHOH DIAL show the water vapor heterogeneity within a range of a few kilometers and its impact on the formation of clouds at the ABL top. The uncertainty of the measured data was assessed by extending a technique, which was formerly applied to vertical time series, to scanning data. Typically, even during daytime, the accuracy of the DIAL measurements is between 0.5 and 0.8 gm$^{-3}$ (or < 6 %) within the ABL, so that now the performance of an RHI scan from the surface to an elevation angle of 90 degrees becomes possible within 10 min. In summer 2014, the UHOH DIAL participated in the Surface-Atmosphere-Boundary-Layer-Exchange (SABLE) campaign in south-western Germany. Volume scans show the water vapor field in three dimensions. In this case, multiple humidity layers were present. Differences in their heights in different directions can be attributed to different surface elevation. With low-elevation scans in the surface layer, the humidity profiles and gradients related to different land use and surface stabilities were also revealed.

## 1   Introduction

Water vapor (WV) is the most important greenhouse gas and plays a key role in Earth's weather and climate, from the surface to the troposphere to the stratosphere. Particularly important are exchange processes between the land surface and the atmospheric boundary layer (ABL) as well as the ABL and the lower troposphere. For example, the diurnal cycles of evapotranspiration, the ABL moisture, and entrainment at the ABL top are the result of feedback processes in the land-atmosphere (LA) system (Seneviratne et al., 2010). However, generally the understanding of LA interaction has been based





on model studies (e.g., Findell et al., 2003; Koster et al., 2006, van Heerwaarden et al., 2009; Santanello et al., 2013) and surface observations but not on suitable data sets including ABL WV fields.

Better parameterizations of land-surface and turbulent transport processes in the ABL are essential for improved weather forecasts (e.g., Ek et al., 2003; Niu et al., 2011, Shin and Hong, 2011, Cohen et al., 2015) and regional climate projections 5 (e.g., Warrach-Sagi et al., 2013; Milovac et al., 2015). These parameterizations were mainly derived by large eddy simulation models (e.g., Mellor and Yamada, 1982; Hong et al., 2006, Hong, 2007; Nakanishi and Niino, 2009; Shin and Hong, 2015) and only to a minor extent by observations. Vertical and horizontal moisture transports via mesoscale circulations and surface heterogeneities can result in convection initiation (CI) as well as the formation of clouds and precipitation (e.g., Behrendt et al., 2011; Corsmeier et al., 2011). All these processes are interacting in a highly nonlinear 10 way so that the WV content needs to be represented very well in weather forecast models (Crook, 1996, Dierer et al., 2009), reanalyses (Bengtsson et al., 2004), and climate models (e.g., Kotlarski et al., 2014).

Models are only as good as the observations, which were used for their parameterization and verification. Advanced observations of WV to study exchange, feedback, and mesoscale circulation processes require the observation of the 3D WV field with a resolution permitting the simultaneous measurement of vertical gradients in the WV distribution in the surface 15 layer, the mixed layer, and the entrainment layer at the ABL top. Only if these gradients are resolved the corresponding transport processes can be studied and parameterized (Monin and Obukhov, 1954; Wulfmeyer et al., 2015b). However, the distribution of ABL WV and its evolution in time is neither fully understood nor sufficiently observed and nor adequately reproduced in weather and climate forecasting models. A detailed overview about these processes and the requirements set to suitable observing systems is given by Wulfmeyer et al. (2015a).

20 For WV measurements passive and active remote sensing instruments as well as in situ sensors are available. In situ sensors deliver only data from one location at one time, thus, remote sensing instruments are preferred for studying the vertical and horizontal WV structure in the ABL. However, many remote sensing systems only provide integrated WV (IWV) data which give no spatial information. Passive instruments like infrared (IR) spectrometers or microwave radiometers (MWRs) are able to retrieve WV profiles based on a first guess with temporal resolutions of 5-10 min. Their vertical range resolutions are 25 100 m for IR or several 100 m for MWR at the land surface and approximately 800 m for IR and 2000 m for MWR at the top of the ABL (Löhnert et al., 2009; Blumberg et al., 2015; Wulfmeyer et al., 2015a). Due to these coarse vertical resolutions, fine structures and gradients cannot be resolved. A combination of several systems may be used to determine horizontal structures with the tomography technique. This was simulated for scanning MWRs (Steinke et al., 2014) but the vertical resolution still remain low due to the coarse resolution of the initial signals and inaccurate knowledge of initial fields. To 30 analyze the aforementioned processes in the ABL, like LA feedback or CI, higher WV resolutions in time and space are needed. The corresponding requirements are summarized in Table 1 in Wulfmeyer et al. (2015a).





Lidar, with its ability to measure WV with high temporal and spatial resolution and high accuracy, has the means to investigate the relevant atmospheric processes. For WV profiling with lidar, the Raman technique (e.g., Melfi et al., 1969; Whiteman et al., 1992; Behrendt et al., 2002; Hammann et al., 2015) or the WV differential absorption lidar (WVDIAL) technique (Schotland, 1966; Browell et al., 1979, Bösenberg, 1998, Wulfmeyer and Bösenberg, 1998) can be applied. The

WV Raman lidar (WVRL) technique uses the inelastic scattering (usually in the ultraviolet (UV)) and detects the Raman shifted signals of WV and nitrogen. The ratio of these two signals is then calibrated to derive the WV mixing ratio. As inelastic backscatter signals are used, the signal intensities are quite low and the maximum range is limited especially in daytime. The WVDIAL technique uses two elastic backscatter signals at wavelengths with high and low absorption of WV. In contrast to WVRL, the WVDIAL technique is self-calibrating and needs no further information than the absorption cross-

section at the used wavelengths (Browell et al., 1979). Due to the elastic backscatter signals, the signal-to-noise ratio (SNR) is much higher for the detected signals. This helps to reaching larger range also during daytime and allows for keeping integration times short.

The horizontal variations of the moisture fields can be detected with scanning lidar. Scanning, solar-blind WV Raman lidar measurements were used, for example, to observe the WV structures in the ABL (Eichinger et al., 1999, Whiteman et al.,

2006, Froidevaux et al., 2013, Matsuda, 2013) or to estimate latent heat fluxes at the surface (Eichinger et al., 2000). Due to the operation in the solar-blind spectrum, the range of this type of WVRL is limited by ozone absorption resulting to a maximum range of several 100 m. Operating the WVRL in the UV but at larger wavelength permitted to extend the range of WVRL to the entire troposphere and to perform WV scans during nighttime (Goldsmith et al., 1998). However, the SNR is still limited during daytime making scans under daylight conditions very difficult.

Therefore, we focused on the WVDIAL technique and developed a scanning system permitting high-resolution scans of the WV field even during daytime. The University of Hohenheim (UHOH) WVDIAL is a ground-based, mobile instrument which demonstrated already vertical measurements in the ABL with high resolution and accuracy (Bhawar et al., 2011; Muppa et al., 2015, Wulfmeyer et al., 2015b). Here, we present different types of scanning measurements of this system and discuss the measurement uncertainties. Particularly, we are demonstrating the measurement of a 3D water-vapor field, which

to our knowledge was achieved for the first time.

We focus on three different scan strategies:

1) With range-height-indicator (RHI) scanning measurements, the vertical WV structure over a certain horizontal range can be observed. We present a new technique to determine the corresponding 2D error field and discuss the measurement performance in this configuration. This is essential to find the best compromise between scan speed, temporal and range

resolution in order to detect the fine structure of the WV field with high confidence.

2) Conical scans can be performed to study the humidity variations from the mixed layer throughout the ABL top. Combining several of such scans with different elevations yields a 3-dimensional image of the ABL moisture field. We



discuss a corresponding first measurement and derive the error statistics in order to characterize fine structures in the WV field.

3) Low elevation scans close to the surface can be used to study LA feedback processes. Also here, the error statistics is derived in order to investigate how accurate the small-scale variability of the ABL WV field from to the surface to the mixed layer can be derived.

The UHOH DIAL system is introduced in section 2. The DIAL technique and the data analysis procedure are presented in section 3. The three different scanning strategies are discussed in sections 4-6. Finally, results are summarized in section 7.

## 2 University of Hohenheim Water Vapor DIAL System

The laser transmitter of the UHOH DIAL (Figure 1) is based on a Ti:Sapphire ring laser (Schiller, 2009; Wagner et al., 2011, 2013) which is end-pumped with the frequency-doubled radiation of a pulsed diode-pumped Nd:YAG laser (Ostermeyer et al., 2005). Frequency-control is realized with the injection seeding technique (Barnes et al., 1993a,b) in combination with a resonance frequency control (e.g., Wulfmeyer et al., 1995, Wulfmeyer and Bösenberg, 1996). For the measurements described in this paper, the previous transmitter configuration (Wagner et al., 2013) had been modified by the following aspects: The pump laser has undergone a full redesign and consists now of an unidirectional ring oscillator in triangle configuration. The resonator of the Ti:Sapphire laser has also been changed. Now, four resonator mirrors form a dynamically-stable ring resonator in bow-tie configuration (Metzendorf et al., 2012). A new seed laser system with two frequency-stabilized distributed feedback (DFB) laser diodes as injection seeders (Späth et al., 2013) was operated for the first time as part of the transmitter during HOPE. For the SABLE campaign, the online DFB laser has later been replaced by an external cavity diode laser (ECDL) with excellent passive frequency stability (Metzendorf et al., 2015) combined with the Drever-Hall-Wulfmeyer frequency control technique for pulsed, injection-seeded laser (US patent #6,633,596, Wulfmeyer et al., 2000).

The UHOH DIAL has two configurations for transmitting the laser pulses to the atmosphere and receiving the backscattered lidar signals: one for vertical pointing measurements and one for scanning measurements. In the scanning configuration (the only one shown in Figure 1b), the laser pulses are coupled into an optical fiber and transmitted to the atmosphere via a 20 cm telescope. The laser output power is currently restricted to 2 W in this configuration because experiments showed that this type of fiber accepts only up to this power level before being damaged. Due to losses when coupling the light into the fiber and out as well as due to the transmitting telescope optics, the power transmitted into the atmosphere is then 1.6 W. The backscattered photons are collected with an 80-cm diameter telescope with a focal length of 10 m. The transmitting and receiving telescopes are mounted together on the scanner unit using a receiver in Coudé configuration with excellent pointing stability on the detector, which allows for 3-dimensional observations. The scanner unit can be operated with speeds between 0.1 to 6° s$^{-1}$. The control software of the scanner unit offers predefined modes like range-height-indicator scans





(RHI – varying elevation angle and fixed azimuth angle) and conical scans (varying azimuth angle and fixed elevation angle) as well as different types of volume scans. The signals are detected by an avalanche photodiode (APD) in combination with a highly linear and low-noise transimpedance receiver package from the German Aerospace Center DLR. Further details of the transmitter-receiver unit are presented in Riede et al. (2012).

The 14-bit data acquisition system (transient recorder MI.4032, Spectrum GmbH, Germany) records the atmospheric backscatter signals with typically 10 MHz sampling rate resulting in a range resolution of the raw signals of 15 m. We store the backscatter signals of each laser shot, which gives us maximum flexibility later when analyzing the data. Together with the lidar signals, elevation and azimuth angles of the telescope are recorded with each pulse. For the WV calculation, the raw data are typically averaged over 1 s to 1 minute in time for the online and offline data. Range averaging is applied within the

WV derivation (so-called Savitzky-Golay derivation). The system specifications of the UHOH DIAL are summarized in Table 1.

In recent years, the performance of the UHOH DIAL system was investigated within several intercomparison campaigns. Bhawar et al. (2011) performed an extensive comparison study between the UHOH DIAL and six other WV lidar systems during the Convective and Orographically-induced Precipitation Study (COPS) in 2007 (Wulfmeyer et al. 2011; Behrendt et

al., 2013). They found a bias of only -1.43 % for the UHOH DIAL relative to the mean of all measurements. In 2013, in Stuttgart-Hohenheim, a further comparison study with Vaisala RS-92 radiosondes was performed and resulted in a mean bias of -1.0 % ± 2.6 % (Späth et al., 2014). Following the method of Lenschow et al. (2000) and Wulfmeyer et al. (2015b), an analysis of the autocovariance function of the WV time series at each height is used to distinguish between instrumental noise and atmospheric variances. This technique yielded a noise error of < 5 % up to 2 km using a time resolution of 1-10 s

(Muppa et al., 2015). Here, we are demonstrating for the first time how this technique can be modified and adapted to perform error analysis of 2D to 3D scanning measurements.

## 3    Data Processing and Derivation of WV Profiles

### 3.1    DIAL methodology

With the DIAL technique the number density of water vapor (or other trace gases like ozone, methane, etc.) can be measured

directly with two backscatter lidar signals. One signal is tuned to a wavelength with strong absorption of WV ($P_{on}(r)$: online signal) and the other signal to a wavelength with weak absorption ($P_{off}(r)$: offline signal). The range $r$ is measured from the lidar system to the scattering volume along the line of sight. The return signals $P(r)$ for each wavelength can be described with the lidar equation of elastic backscattering as (Wulfmeyer et al. 2015a)





$$P_{v_0}(r) = P_0 \eta \frac{c\Delta t}{2} \frac{A_{\text{tel}}}{r^2} O(r) \Gamma_{\text{air},v_0}^2(r) \cdot$$

$$\cdot \left\{ \beta_{\text{par},v_0}(r) \int_{-\infty}^{+\infty} S_{\text{L}}(v-v_0) \Gamma_{\text{WV}}^2(v,r) F_R(v,r) \mathrm{d}v + \right.$$

$$\left. + \beta_{\text{mol},v_0}(r) \int_{-\infty}^{+\infty} \left[\left[ S_{\text{L}}(v-v_0) \Gamma_{\text{WV}}(v,r) \right] \times DB(v,r) \right] \Gamma_{\text{WV}}(v,r) F_{\text{R}}(v,r) \mathrm{d}v \right\} + P_{\text{B}} \tag{1}$$

with the initial intensity $P_0$ at the laser frequency $v_0$, the system efficiency $\eta$, the speed of light $c$, the time resolution of the system $\Delta t$, the telescope area $A_{\text{tel}}$, the overlap function $O(r)$, the transmission of the atmosphere $\Gamma(r)$ as

$$\Gamma_v(r) = \Gamma_{\text{air},v}(r) \ \Gamma_{\text{WV}}(v,r) = \exp\left\{ -\int_0^r \left[ \alpha_{\text{par},v_0}(r') + \alpha_{\text{mol},v_0}(r') \right] \mathrm{d}r' \right\} \exp\left\{ -\int_0^r \alpha_{\text{WV}}(v,r') \mathrm{d}r' \right\} \ , \tag{2}$$

the particle and molecular extinction coefficient $\alpha_{\text{par},v}(r)$ and $\alpha_{\text{mol},v}(r)$, which are only slightly dependent on frequency $v$, the extinction coefficient of water vapor (of the trace gas) $\alpha_{\text{WV}}(v,r)$, the particle and molecular backscatter coefficient $\beta_{\text{par},v}(r)$ and $\beta_{\text{mol},v}(r)$, the normalized laser spectrum at the ground $S_{\text{L}}$, the spectral broadening due to Rayleigh scattering $DB$, the transmission function of the receiver interference filter $F_{\text{R}}$, and $P_{\text{B}}$ the background signal. $\alpha_{\text{WV}}$ is related to the absorption cross-section $\sigma_{\text{WV}}$ and the WV number density $N_{\text{WV}}$ by

$$\alpha_{\text{WV}}(v,r) = N_{\text{WV}}(r) \ \sigma_{\text{WV}}(v,r) \ . \tag{3}$$

Our laser is designed so that the laser spectrum can be considered a delta distribution. In this case, the derivation of the WV profile becomes independent of any laser parameters, which is called narrow-band DIAL. Furthermore, we assume that the overlap function is either the same for online and offline signals or height independent as well as we consider the interference filter transmission function is constant over the frequency range of interest and not dependent on range. Then,

calculating the ratio of Eq. (1) for online and offline wavelengths, applying the relation of Eq. (3), and solving for the WV number density $N_{\text{WV}}$ leads to the narrow-band DIAL equation

$$N_{\text{WV}}(r) = \frac{1}{2(\sigma_{\text{on}}(r) - \sigma_{\text{off}}(r))} \frac{\mathrm{d}}{\mathrm{d}r} \ln\left( \frac{P_{\text{off}}(r) - P_{\text{B,off}}}{P_{\text{on}}(r) - P_{\text{B,on}}} \right) +$$

$$+ \frac{1}{2(\sigma_{\text{on}}(r) - \sigma_{\text{off}}(r))} \frac{\mathrm{d}}{\mathrm{d}r} \ln\left( \frac{\beta_{\text{par,on}}(r) + \beta_{\text{mol,on}}(r) \Gamma_{\text{WV,on}}^{-1}(r) \int_{-\infty}^{\infty} \Gamma_{\text{WV}}(v,r) DB(v-v_{\text{on}},r) \mathrm{d}v}{\beta_{\text{par,off}}(r) + \beta_{\text{mol,off}}(r)} \right) , \tag{4}$$



the index 'on' and 'off' implies that the specific variable is taken at the online or offline wavelength, respectively. Online and offline wavelengths are chosen close to each other because then the particle and molecular extinction and backscatter coefficients for online and offline wavelengths cancel when taking the ratio of the signals in Eq. (4). All system parameters which are constant with range $r$ cancel because of the derivative. Thus, for DIAL measurements no calibration is needed .

Only the values of the absorption cross-sections at online and offline wavelength $\sigma_{on}$ and $\sigma_{off}$ have to be known were accurately.

The WV cross section $\sigma_{WV}(v, T, p, N_{WV})$ depends on $N_{WV}$ by self broadening, temperature $T$ and air pressure $p$ at a certain frequency $v$. These dependencies have been measured very accurately in the laboratory and collected in databases, e.g., the HIgh-resolution TRANsmission molecular absorption database (HITRAN) (Rothman et al., 2013). Selecting specific

absorption lines with low ground-state energy, it was shown that the dependence of the cross sections on the atmospheric temperature, pressure, and WV profiles is weak and that it is sufficient to assume mean hydrostatic and adiabatic conditions merely using surface values. This makes DIAL the most accurate WV remote sensing technique to date.

Suitable wavelength regions for WV DIAL were studying over a large wavelength range by Wulfmeyer and Walther (2001a,b). Specific DIAL systems were developed, e.g., near 720 nm (Bruneau et al., 2001; Wulfmeyer and Bösenberg

1998), near 820 nm (Ismail and Browell, 1989; Ertel, 2004; Schiller et al., 2007; Vogelmann et al., 2008; Behrendt et al. 2009), near 935 nm (Machol et al., 2004, 2006; Wirth et al., 2009; Fix et al., 2011), and near 1480 nm (Petrova-Mayor et al., 2008). The UHOH DIAL operates at wavelengths near 818 nm because this wavelength region can be reached well with a Ti:Sapphire transmitter and offers a sufficiently large range of WV absorption cross sections (Wagner et al., 2011, 2013).

The second term in Eq. (4) describes the Rayleigh-Doppler correction term related to the broadening effect of the laser

spectrum by Rayleigh scattering. The particle backscatter coefficient can be calculated from the offline signal (Fernald et al., 1972; Fernald, 1984). Ansmann and Bösenberg (1987) showed that this correction becomes significant when strong particle backscatter gradients are present. However, they only considered that the online laser wavelength is at the peak of the water vapor absorption line.

Within the analyses presented here this effect was not critical as confirmed not only by comparisons with radiosoundings but

also with new sensitivity analyses considering a frequency agile operation of our laser transmitter. This reduced sensitivity to the Rayleight-Doppler correction was due to two reasons. Firstly, most of the sampled air masses were located within the ABL where no large particle backscatter gradients were present. Secondly, the Rayleigh-Doppler effect is strongly reduced if the online frequency is located on the wing of the absorption line. In this case, the integral in Eq. (4) becomes approximately $\Gamma_{wv}$ so that the nominator and the denominator of the term cancel – independent of the aerosol gradient. More details can be

found in Späth et al. (2015). For the measurements discussed here, we selected an online frequency away from the peak absorption but still strong enough to produce sufficient differential absorption. This selection allowed us to optimize the

sensitivity of the DIAL measurements in the range of interest for the moisture values present (Späth et al., 2014). Thus, the so-called Schotland approximation of the DIAL equation (Schotland, 1966, 1974)

$$N_{\mathrm{WV}}(r) = \frac{1}{2(\sigma_{\mathrm{on}}(r) - \sigma_{\mathrm{off}}(r))} \frac{\mathrm{d}}{\mathrm{d}r} \ln\left(\frac{P_{\mathrm{off}}(r) - P_{\mathrm{B,off}}}{P_{\mathrm{on}}(r) - P_{\mathrm{B,on}}}\right) \tag{5}$$

could be used for the cases presented here. In the following, the derived moisture values in number density are transformed in absolute humidity $\rho$ in units of $\mathrm{gm^{-3}}$.

## 3.2 Data Processing

In case of the UHOH DIAL, the atmospheric backscatter data are recorded for each laser shot. Later, these data are averaged in time over typically 1 to 10 s and background corrected. The absorption cross-section profiles for the selected online and offline wavelengths are determined using profiles of temperature $T(z)$ using a surface value in combination with the temperature gradient from the US standard atmosphere, a hydrostatic pressure $p(z)$ also initialized with a surface value and an atmospheric mean temperature in the range of interest. As well as an initial guess for the water vapor number density $N_{\mathrm{WV}}(z)$ from the U.S. standard atmosphere (NASA, 1976). The spectroscopic parameters of water vapor we take from the latest compilation of HITRAN described by Rothman et al. (2013).

The water vapor profile is then calculated according to Eq. (5). The Savitzky-Golay (SaGo) method is applied for the derivative with respect to range $r$ (Savitzky and Golay, 1964). This method calculates the first derivative using a certain number of data points. The resulting range resolution $\Delta R$ is approximately half of the SaGo window length, as the weighting function is parabolic (Ehret et al, 2001). Typically our SaGo window range consists of 4 data points on each side of a specific range resulting in a range resolution of $9 \times 15\mathrm{m}/2 \cong 67.5\mathrm{m}$. The step size of the data points used for the derivation of the WV profile is kept with 15 m. Depending on whether Eq. (4) or (5) must be applied, either an iteration is necessary to derive the WV profiles (Eq. 4) or a direct derivation is sufficient (Eq. 5). However, in both cases the resulting solution is unique and even if an iteration must applied to find the result it converges very quickly after 1-3 iterations. Thus, the DIAL technique *derives* WV profiles in contrast to passive remote sensing where WV profiles are *retrieved* based on a first-guess profile. The whole chain of data processing is summarized in a flow chart (see Figure 2).

## 3.3 Analysis of scanning data

Up to this step, the analysis procedure is similar for vertical and scanning measurements. After calculating WV of scanning measurement, the data can be plotted and used for further analysis in polar coordinates $(r, \Theta, \varphi)$. Alternatively, the data can be gridded to a regular horizontally and vertically spaced grid $(x, y, z)$ (see also Figure 2). We prefer gridded data because





atmospheric variations are usually horizontally or vertically oriented. 3D data sets can be analyzed by extracting slices of different orientation.

To estimate instrumental noise $\Delta\rho$ of vertical measurements, we apply the method of Lenschow et al. (2000) and Wulfmeyer et al. (2015b). Here, the autocovariance function (ACF) of the humidity fluctuations for one range bin is determined. The

ACF at lag 0 gives the total variance which is the sum of atmospheric variance and noise variance. The atmospheric variance can be separated from the instrumental noise by extrapolating the ACF to lag 0. For conical scanning measurements, this approach can be used without further modification because data points of a certain range are at the same height.

For scanning measurements in RHI mode, the determination of the atmospheric variance is more complicated. When a time series with a large number of fast RHI measurements with small periods between consecutive scans is available, one can just

use the time series of data at one range and elevation. But also the noise within single RHI data can be determined. In contrast to conical scans, rings of constant range cover different heights for RHI data and thus clearly different atmospheric variance values. But even more importantly, the instrumental noise within an RHI scan that covers a large part of the ABL differs for fixed range because the humidity and thus optical depth show large differences (Wulfmeyer and Walther, 2001b). In consequence, one has to group the measured RHI data then in a more sophisticated way. In the following, we suggest such

an approach.

We have tested several different concepts for grouping the data. A simple 1-dimensional approach is to take a number of range bins from one slant profile. But as the required number of independent measurement points is about 10, this still leads to very similar problems as discussed above when selecting measurement points of constant range. Thus, we finally decided for grouping the measured data 2-dimensionally. We calculate the noise estimation with 3 independent data points of 3

profiles giving $3 \times 3 = 9$ data points for each group that is analyzed. The vertical noise profile is then obtained from groups of which the central data points are at a certain horizontal distance to the lidar (Figure 3).

When using RHI data sets with very high spatial resolution (in range and elevation), the instrumental noise is much larger than the atmospheric variance. Thus the sequence of the 9 data points of each group used for the ACF analysis becomes irrelevant and one can just use lag 0 as upper limit for the instrumental noise estimation. Finally, the resulting noise values

are scaled to the temporal and spatial resolution of the averaged and gridded data according to (Ismail and Browell, 1989)

$$\frac{\Delta\rho}{\rho} \propto \left(\Delta t\right)^{-0.5}\left(\Delta r\right)^{-1.5} . \tag{6}$$

In the following, we present several examples of 2D and 3D scans of the WV fields, analyse the results, and apply our new tools for error analyses.



## 4 Range-Height-Indicator Scans for the Investigation of 2D Turbulence in the ABL and of Clouds Measured During HOPE

### 4.1 Instrumental setup

To capture the horizontal and vertical WV field and its relation to 2D ABL turbulence statistics and cloud formation, RHI
scanning measurements are preferable. With vertical measurements only observations in the so-called Eulerian specification are possible which means that the atmosphere is observed while advecting through the lidar beam. Here, temporal and spatial changes of the measured data are entangled. With RHI scans, this is not the case (or at least much less) so that both temporal and spatial differences of the moisture field in the ABL can be studied.

The UHOH DIAL was operated in RHI mode within the "HD(CP)$^2$ Observation Prototype Experiment" (HOPE) near
Forschungszentrum Jülich, in western Germany (see http://www.hdcp2.eu/Campaign-HOPE.2306.0.html). The aim of the experiment was to produce a dataset of atmospheric measurements for the investigation of land-atmosphere interaction, cloud formation, aerosol-cloud-microphysics as well as weather and climate model evaluation at the 100-m scale. For HOPE, three supersites were set up in a triangular configuration with distances of about 4 km between each other. All these sites were equipped with in situ and remote sensing instruments to measure atmospheric parameters. With the different
instruments, the temporal and spatial heterogeneity of the convective boundary layer (CBL) was investigated concerning WV, temperature, and wind fields as well as the distribution of aerosol particles and clouds. The UHOH DIAL was located at the HOPE supersite near Hambach (50°53'50.56" N, 6°27'50.39" E, 110 m above sea level). At the same site with the UHOH DIAL, the UHOH Rotational Raman Lidar (RRL) (Radlach et al., 2008; Hammann et al., 2015) measured temperature and the KITcube (Kalthoff et al., 2013) observed - among others - the wind field with scanning Doppler lidar
systems (Träumner, 2010) and the surface energy balance (Kalthoff et al., 2006; Krauss et al., 2010). The UHOH DIAL provided measurements of more than 180 h on 18 intensive observation periods (IOPs) in vertical and different scanning modes. The high-resolution fields of the measured thermodynamic variables are also being used to derive higher-order moments of turbulent fluctuations (Muppa et al., 2015; Behrendt et al., 2015a) as well as sensible and latent heat fluxes (Wulfmeyer et al., 2014; Behrendt et al., 2015b). These results will be used for the verification of current approaches of
turbulence parameterizations as well as of the development and tests of new turbulence parameterizations. Further details are found in Wulfmeyer et al. (2015b).

### 4.2 Performance and analyses of RHI scans

Our measurements were performed during IOP 4 on 20 April 2013. On this day, the HOPE domain was under the influence of a high pressure system located with its center over the Baltic Sea. The main wind direction was northeast to east as
confirmed by the radiosoudings. Figure 4 shows temperature, humidity and wind velocity profiles of the radiosonde launched at 07:00 UTC at the DIAL site. The horizontal wind speed within the ABL was between 8 and 10 ms$^{-1}$ with a dip





down to a wind speed of 5.5 ms$^{-1}$ at a height of 1250 m. Temperature and absolute humidity were quite low on this day with ground values of only 5° C and 4.5 gm$^{-3}$, respectively. Between 06:00 and 07:00 UTC, only thin cirrus clouds were present at heights between 7 and 8 km. The surface temperature profile shows that a very shallow unstable surface layer started to develop by surface heating but it was not deeper than 200 m. Consequently, all layers above could be considered as residual

layers. The complex vertical layering is particularly visible in the absolute humidity profile, which shows a series of moisture layers with a thickness of 100 to 200 m, up to 1.6 km.

RHI scanning measurements of the humidity filed in the HOPE region were performed on 20 April 2013 between 06:03 and 07:24 UTC. The time resolution for this analysis is 10 s which results with a scan speed of 0.15° s$^{-1}$ in an angle resolution of 1.5°. As each of these RHI scans took 566 s, an elevation angle range of 85° was covered. Due to the longer path of the laser

beam through the aerosol-loaded air in lower elevation and the corresponding higher extinction as well as due to the higher moisture in the boundary layer and the corresponding stronger attenuation of the online signal, we averaged the data according to the following procedure differently for different ranges. We calculated the absolute humidity with different range resolutions $\Delta R$ to keep the angle resolution. Afterwards, the radial data have been gridded to a horizontal-vertical grid with a resolution of 50 m. Finally, the data with different resolutions were merged according to the horizontal distance: up to

a distance of 1.3 km $\Delta R$ = 142.5 m, up to 2.5 km $\Delta R$ = 307.5 m, up to 3.0 km $\Delta R$ = 457.5 m, and up to 4.2 km $\Delta R$ = 997.5 m. A noise error analysis was carried out as described in Sect. 3 and profiles for the horizontal distances of 1.3, 2, 3 and 4 km are shown in Figure 5. For all distances, the error profiles show a clear increase at the top of the ABL which rises from an altitude of around 0.9 km at 1.3 km distance to 1.1 km at 4 km distance. Below the top of the ABL, the noise values are below or around 0.2 gm$^{-3}$ and above the ABL top the noise reaches values between 0.5 and 0.8 gm$^{-3}$. This results in an upper

limit of a relative noise error of < 6 % within the ABL. The noise profile for 1.3 km distance does not show the smallest values but this is maybe due to the assumption of neglecting atmospheric fluctuations which become relatively larger when the instrumental noise decreases.

Figure 6 shows scanning humidity measurements in the HOPE region on 20 April 2013 between 06:03 and 07:24 UTC. The measurements were performed towards the two other experimental sites of the HOPE campaign: towards the Leipzig

Aerosol and Cloud Research Observations System (LACROS) southwards and towards the Jülich ObservatorY for Cloud Evaluation (JOYCE) south-westwards. The plots are geo-located to the Earth surface and cover a horizontal range of 0.7 to 4.2 km up to an altitude of 2 km.

The WV field in the two scanning directions showed several similarities but also significant differences revealing the heterogeneities of the ABL in the region. In both directions, three moist layers with drier layers in between can be identified.

The altitudes of the layers differ significantly. Towards LACROS, the moist layers were at 500, 1100 and 1500 m while towards JOYCE they were at 300, 1000 and 1400 m for the measurements for the first two scans between 06:03 and



06:27 UTC (Figure 6a). One hour later (Figure 6b), the measurements show the same number of layers but the height of the lowest one increased to 600 m and 500 m for LACROS and JOYCE direction, respectively.

In general, we see that the ABL was moister in the direction of LACROS than in the direction of JOYCE. In order to illustrate this, we have averaged the scanning data horizontally. The averaged vertical humidity profiles of the four RHI

measurements of Figure 6 are shown in Figure 7. In the profiles, the layer structure discussed above can be identified. The humidity profile of the radiosonde at 07:00 UTC does not fit with an averaged DIAL profile and shows also different structures but the profile stays within the variability of the DIAL measurement towards LACROS. However, different to the DIAL measurements, a radiosonde measures only along its flight path and, thus, can only sample a snapshot of the atmospheric constitution. The horizontal variability within a certain range can also only be determined with scanning DIAL

measurements. For that reasons, horizontally averaged profiles of scanning DIAL measurements are better suited for comparisons with model simulation outputs which give also profiles representative for a whole model grid box (Milovac et al., 2015).

Interestingly, in one of the scans (Figure 6b towards JOYCE) clouds appear while all others are cloud-free. Two clouds can be identified at 500 m altitude at the top of the lowest moist layer. The large extinction of the clouds prohibits measurements

inside the clouds and beyond. At 560 m altitude the profiles of the radiosonde (Figure 4) show a zero-crossing in the temperature and a relative humidity of only about 80%. The DIAL measurements show values of about 4.5 and 5 $\text{gm}^{-3}$ below the clouds at 1.5 and 3 km distance, respectively. As at least 100% relative humidity is needed for cloud formation, this corresponds to a required absolute humidity of 4.8 $\text{gm}^{-3}$ at a temperature of 0° C. Clearly, the observed clouds are related to locally higher moisture values as revealed by the DIAL scans, which is also seen in Figure 7d by an upper limit of the 1$\sigma$

standard deviation above 5 $\text{gm}^{-3}$ in heights between 300 and 700 m. The measurement one hour before showed already these humidity values at around 250 m altitude which was not sufficient to reach saturation but indicate that the humidity came from the ground and reached the condensation level during the last scan.

## 5 Volume Scans for the Investigation of the 3-Dimensional Water Vapor Field

### 5.1 Instrumental set up

With volume scans, the relation of the moisture field to surface properties can be investigated more in detail. The observation of 3D humidity is either possible by a series of fast RHI scans using different azimuth angles or by a series of continuous 360° scans with different elevation angles.

For the first time with the UHOH DIAL, the latter configuration was applied during the Surface Atmosphere Boundary Layer Exchange (SABLE) field campaign in August 2014. The SABLE campaign took place near Pforzheim

(48°55'45.85"N, 8°42'19.57"E, 320 m above sea level) in the Black Forest (south-west Germany) as part of the Research Unit 1695 "Regional Climate Change" of the German Research Foundation (DFG, see https://klimawandel.uni-





hohenheim.de/start?&L=1). The UHOH DIAL was collocated with the UHOH RRL for temperature measurements and with three Doppler lidar systems for measuring the wind velocities as well as with a synergy of surface in situ sensors distributed in the fields, e.g., eddy covariance stations (Wizemann et al., 2015). The results of these campaigns shall contribute to an improved understanding of the relations between surface properties and boundary layer characteristics, shallow cumulus convection, as well as convection initiation.

During the special observations period (SOP) 2 on 22 August 2014, the 3-dimensional WV field was observed with a volume scan. On this day, a low pressure system over Scandinavia and a high pressure system over Eastern Europe provoked westerly flow in the SABLE domain. Stratus clouds occurred over the measurement site with a bottom height of about 2.5 km height. These clouds reduced the surface heating so that no convective boundary layer was forming. Consequently, we can assume that the structures of the moisture field were largely advected and modified locally mainly by orography. The radiosonde from 09:30 UTC showed at the ground a temperature of 16° C and a relative humidity of 60 %. The wind was calm ($< 1.5$ ms$^{-1}$) at the ground up to 300 m and increased then to 6 ms$^{-1}$ at 600 m in east-northeast direction.

## 5.2    Performance and analyses of the volume scan

The area around the DIAL site was observed performing a series of 360° conical scans around the vertical with elevation angles of 50°, 60°, 70°, 80°, and 90°. Figure 8 shows a 3D view of the 50° cone of the volume scan between 08:40 and 09:40 UTC. For this measurement, the scan speed was 0.4° s$^{-1}$ resulting in a total duration of 15 min per cone. Consequently, the full volume with 5 cones was scanned within 75 min. The start and the end directions were southward oriented and the scans moved for the 50°, 70°, and 90° cone clockwise and counter-clockwise for the other scans of 60° and 80°. The WV calculation was performed with 10 s averaged profiles and a 67.5 m range resolution. For plotting, the WV data were transferred from the polar coordinates to Cartesian coordinates and each profile was expanded over an azimuth range of 4°.

The instrumental noise was determined with the method of Lenschow et al. (2000) for each elevation angle separately. Profiles of the absolute and relative noise are shown in Figure 9. Above 400 km, the noise level increases with height. For lower elevation angles the noise increases stronger with respect to altitude because of larger range values. The profile of the lowest elevation angle shows a maximum noise of $< 0.35$ gm$^{-3}$ or $< 7\%$ at 750 m altitude. Below 400 m, the noise also increases, but here due to overlap effects. However, this does not occur in the absolute humidity data as the noise is $< 0.8$ gm$^{-3}$.

The measurement in Figure 8 shows two moist layers. The lower layer reached altitudes up to between 300 and 400 m with humidity values of 7 to 8 gm$^{-3}$. This layer was topped by a drier layer with 5.6 gm$^{-3}$ and a second moist layer at 600 m altitude with a humidity of about 6.5 gm$^{-3}$.

To get an insight to the whole volume, Figure 10a shows cross-sections and illustrates the whole data set of the volume scan 3-dimensionally. In addition and for orientation, the cutting planes are depicted of which the water vapor distribution is then



shown in Figure 10b-i. The figure contains three vertical cross-sections in north-south (b-d) and three in west-east direction (e-g) as well as horizontal cross-section planes at two height levels (h-i). The vertical cross-section images depict the vertical structure at different distances to the DIAL similar to what was shown above with RHI scans. But from the volume scan the cross-sections can be extracted of course from different distances and directions of interest. In Figure 10b-g the vertical planes are positioned at +/- 0.2 km and 0.0 km distance with respect to the vertical line above the DIAL location. Also the horizontal cross-section plane can of course be placed at any height of interest. These plots show the WV distribution with respect to the azimuth angle but in contrast to conical scan plots not the data along the line of sight but at one height the data of all conical scans of different elevation angles of the volume scan. The cross-section images depict also the moist layers in the two lower elevation angle scans.

The heights of the layers in Figure 10 are almost similar for all directions. Because of the full cloud cover on this day, convection was very weak and no large eddies were initiated. Thus, the moist layer from the ground grew slowly and variations in the layer height correspond to the surface elevation. Here, in south direction at a distance of 0.6 km from the DIAL a small hill was located with an increase of the surface elevation of about 40 m. The measurements of the lowest elevation angle in Figure 10b-d show indeed a higher altitude of the boundary layer of about 50 m at a distance of 0.5 km southwards. Also to the east a small trend to higher altitudes of the boundary layer top can be observed. This is also indicated by the horizontal plane image in Figure 10i for the south-easterly direction and a distance of 0.6 km. This area corresponds in our case to the lee side of the small hill so that the higher moisture might be explained by a modification of the moisture field by shifted overflow lifting.

Averaging the volume data horizontally provides the mean humidity profile. The profile was plotted in Figure 11 up to an altitude of 0.8 km. The thin lines mark the horizontal WV variability similar to Figure 7. The radiosonde profile at 09:30 UTC is given in the diagram as well. Again, there were differences between DIAL and radiosonde measurements but the radiosonde captured similar moisture layers and stayed almost within the variability range of the DIAL profile.

## 6 Low-Elevation Range-Height-Indicator Scans for the Investigation of the Surface Layer

### 6.1 Instrumental set up

On 12 August 2014 the SABLE domain was under westerly flow due to low pressure systems located over the Baltic Sea and Scandinavia; one day before a cold front passed through the measurement area. Vertical DIAL measurements before and after the scanning measurements (not shown here) indicate a CBL height of between 1.0 and 1.5 km and a residual layer up to 2 km. The WV content in these two layers were up to 8 $gm^{-3}$ and around 5 $gm^{-3}$, respectively. During the scanning period, cirrus clouds at 8 km altitude were present and also few low level clouds at 2 to 2.5 km altitude at the top of the CBL but these clouds did not inhibit large surface fluxes (sensible heat flux 100 $Wm^{-2}$ and latent heat flux 240 $Wm^{-2}$) and the development of a CBL. The radiosonde profiles from 10:30 UTC measured a temperature of about 18° C and a relative



humidity of 60 % at the ground. The relative humidity increased with height and reached 80 % at the top of the CBL. The wind speed was low with 4-5 ms$^{-1}$ at the surface and increased linearly with height; the wind direction was West.

In order to observe the surface layer, measurements are needed which reach as close as possible the land surface or the canopy level. Due to incomplete overlap of the outgoing laser beam and the field-of-view of the receiver in the near range, vertical measurements of the UHOH DIAL start only a few hundred meters above ground. With low-level RHI scans, these low-level measurements can be realized. Furthermore, the variations in the humidity structures can be related to different types of land use along the line of sight of the low-level scanning measurements.

During the SABLE campaign (see Sect. 4), these low-level scanning measurements were performed to investigate the properties of the atmospheric surface layer. The low elevation scans covered angles between 0 and 12° above horizon. In order to reach a high vertical resolution, the scan speed was 0.2° s$^{-1}$, which resulted in a time duration of 1 min per scan. In Figure 12 the averaged WV field of IOP 4 on 12 August 2014 of the measurement period between 11:00 and 12:00 UTC is shown.

## 6.2 Performance and analyses of the low-level RHI scans

For the WV calculation, 1 s averaged profiles and a SaGo window length of 135 m were used. All scans of the one hour period (60 scans) were averaged with 1° angle resolution resulting in an final time resolution of 260 s. Then, the data were gridded to an x-y grid with a resolution of 50 m x 10 m. With the high spatial resolution, small variations in the absolute humidity values (notice color scale 9.4 to 10.4 gm$^{-3}$ in Figure 12) at different heights and distances can be identified.

The instrumental noise for the low elevation scan was estimated with the ACF method. The data were used in the initial radial polar coordinates and data points of all scans at a certain range bin covering a 1° angle range were selected for the ACF calculation. The resulting errors were then scaled with Eq. (6) from the 1 s time resolution. In Figure 13, the noise errors were plotted as error bars with the absolute humidity profiles at the distances of 400, 800, and 1200 m, respectively. All three profiles show a constant noise level for the whole profile which can be expected for the small covered height range because all data points of one profile belong to a similar range bin and as the profile stays within in the boundary layer the optical thickness is constant over the whole height range. Of course, the noise level increases with distance but the noise value stays lower than 0.3 gm$^{-3}$ for 400 m, lower than 0.4 gm$^{-3}$ for 800 m and lower than 0.9 gm$^{-3}$ for 1200 m. These values translate to relative values of less than 0.3 %, less than 0.4 % and less than 1 %, respectively.

The humidity values close to the ground are higher than above and at 1200 m distance the humidity was higher than in 600 m distance. Because these measurements are close to the ground, it is possible to relate these changes in horizontal direction to the vegetation at the ground. For the measurement in Figure 12, the vegetation can be separated in three categories: In the near range up to 450 m there was a maize field, up to 1050 m the ground was covered with grassland, and farer away we scanned over forest. The terrain profile was flat for the maize field and slightly uphill for the grassland while





the forest was located on a small hill reaching an altitude of 375 m above sea level in 1300 m distance. The measurement in Figure 12 shows that there was more water vapor in the atmosphere above the maize field and above the forest which was likely due to higher evapotranspiration than above the grassland.

This one hour mean profile close to the ground is similar to what was measured by Eichinger et al. (2000) with a scanning Raman lidar. However, with the WVDIAL technique a larger range can be investigated. In the future, such data can be used to estimate the spatial distribution of the latent heat flux over different kinds of vegetations (Wulfmeyer et al., 2014). For this purpose, the Monin-Obukhov similarity theory (MOST) (Monin and Obukhov, 1954; Brutsaert, 1982) can be applied using the slope of such a vertical humidity profile and simultaneously obtained friction velocity $u^*$.

## 7    Summary

The measurements of the spatial distribution of water vapor by the UHOH DIAL in three different scanning modes were presented. The UHOH DIAL uses a frequency-stabilized Ti:Sapphire resonator as laser transmitter and emits laser pulses at 820 nm. The output power for scanning measurements is currently limited to 1.6 W. The 20 cm transmitting and 80 cm receiving telescopes form the scanner unit which allows scanning measurements of the whole hemisphere (180° elevation, 360° azimuth) with scan speeds between 0.1 and 6° s$^{-1}$. For data analyses typical range and temporal resolutions of 50 to 300 m and 1 to 10 s, respectively, are used. A new method to determine the noise level of scanning measurements was developed and show uncertainties of < 7 % within the ABL. With the DIAL technique it is now possible to determine 3D WV fields with high temporal and spatial resolution including a specific analysis of noise error fields. Therefore, the significance of WV structures in these 2D and 3D fields can be studied and specified in great detail.

Scanning measurements in RHI mode were performed in two directions during HOPE with elevation angles from 5 to 90° up to a horizontal distance of 4 km. With these scans the humidity field was investigated regarding turbulent and mesoscale variability as well as cloud formation. Similar layers for both directions but also differences in altitudes of the layers or in the WV content were observed in the WV field. Four scans depict the evolution of the layers within 90 minutes. In the last measurement of the series, clouds appear at the top of the lowest moist layer where the conditions of 100 % relative humidity for cloud formation were locally fulfilled. Horizontally averaged vertical profiles show also a higher humidity variability for that measurement. The noise analyses show a strong increase at the ABL top and a noise error of < 6 % within the ABL.

For the first time, a volume scan performed during the SABLE campaign presents the 3-dimensional spatial WV distribution within a distance range of 0.8 km around the DIAL. The data show two moist layers with some variations in height for different directions. These variations can be related to variations in the surface elevation, e.g., in southeast direction a small hill with a slightly higher elevation was located. The instrumental noise for this case was calculated to < 0.5 gm$^{-3}$ or < 7 %.

Low elevation scanning measurements obtained humidity structures close to the ground. The presented data were averaged over one hour of scanning measurements and cover a height range from 20-140 m. The horizontal variation of the WV field



can be related to the heterogeneity of the vegetation at the ground. In future work, this kind of measurement can be used to estimate evaporation above different vegetation using the Monin-Obukhov similarity theory. The error for this kind of measurements were estimated to $< 0.3$ gm$^{-3}$ or $< 0.3$ % at 400 m and $< 0.9$ gm$^{-3}$ or $< 1$ % at 1200 m horizontal distance throughout the measured height range.

5  In conclusion, all scanning modes are applicable to observe the spatial distribution of water vapor in the lower atmosphere. Depending on the focus of the research, the scan pattern can be adapted regarding the covered elevation and azimuth angle ranges.

**Acknowledgements**

10  The HOPE campaign was funded by the German Research Ministry under the project number 01LK1212A.

The SABLE campaign was included in the Research Unit 1695 "Regional Climate Change" which was funded by the German Research Foundation (DFG) under the DFG Integrated Project PAK 346, project number WU 356-1 AOBJ: 591757.

**List of Acronyms**

| | | |
|---|---|---|
| 15 | ABL | atmospheric boundary layer |
| | ACF | autocovariance function |
| | APD | avalanche photodiode |
| | BR | beam reducer |
| | CBL | convective boundary layer |
| 20 | CI | convection initiation |
| | COPS | Convective and Orographically-induced Precipitation Study |
| | DFB | distributed feedback |
| | DIAL | differential absorption lidar |
| | DLR | German Aerospace Center (Deutsches Zentrum für Luft- und Raumfahrt) |
| 25 | ECDL | external cavity diode laser |
| | FC | fiber coupler |
| | HD(CP)$^2$ | High Definition of Clouds and Precipitation for advancing Climate Prediction |
| | HITRAN | HIgh-resolution TRANsmission molecular absorption database |
| | HOPE | HD(CP)$^2$ Observational Prototype Experiment |



| | | |
|---|---|---|
| | HR | high-reflection mirror |
| | IOP | intensive observation period |
| | IR | infrared |
| | IWV | integrated water vapor |
| 5 | JOYCE | Jülich ObservatorY for Cloud Evaluation |
| | LA | land-atmosphere |
| | LACROS | Leipzig Aerosol and Cloud Research Observations System |
| | MOST | Monin-Obukhov similarity theory |
| | MWR | microwave radiometer |
| 10 | NASA | National Aeronautics and Space Administration |
| | PD | photo diode |
| | PM | primary mirror |
| | RHI | range-height indicator |
| | RRL | rotational Raman lidar |
| 15 | SABLE | Surface-Atmosphere-Boundary-Layer-Exchange |
| | SaGo | Savitzky-Golay |
| | SM | secondary mirror |
| | SNR | signal-to-noise ratio |
| | SOP | special observations period |
| 20 | TM | transmitting telescope mirror |
| | UHOH | University of Hohenheim |
| | UV | ultraviolet |
| | WV | water vapor |
| | WVDIAL | WV differential absorption lidar |
| 25 | WVRL | WV Raman lidar |



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





Table 1 Instrument specifications of the UHOH DIAL for scanning WV measurements.

| Parameter | |
|---|---|
| Pump power (532 nm) | 14 W |
| Repetition rate | 250 Hz |
| Ti:Sapphire output power (820 nm) | 2 W |
| Pulse energy | 8 mJ |
| Pulse duration | 60 ns |
| Frequency switching | Shot-to-shot on-/offline 2x1 optical fiber switch |
| Wavelength range | 817.7 – 819.0 nm |
| Output power sent into the atmosphere | 1.6 W |
| Transmitting telescope diameter | 20 cm |
| Receiving telescope diameter | 80 cm |
| Scan speed | 0.1 - 6.0° s$^{-1}$ |
| Sampling rate | 10 MHz |
| typical time resolutions | 1 s - 1 minute |
| typical range resolutions | 30 - 300 m |



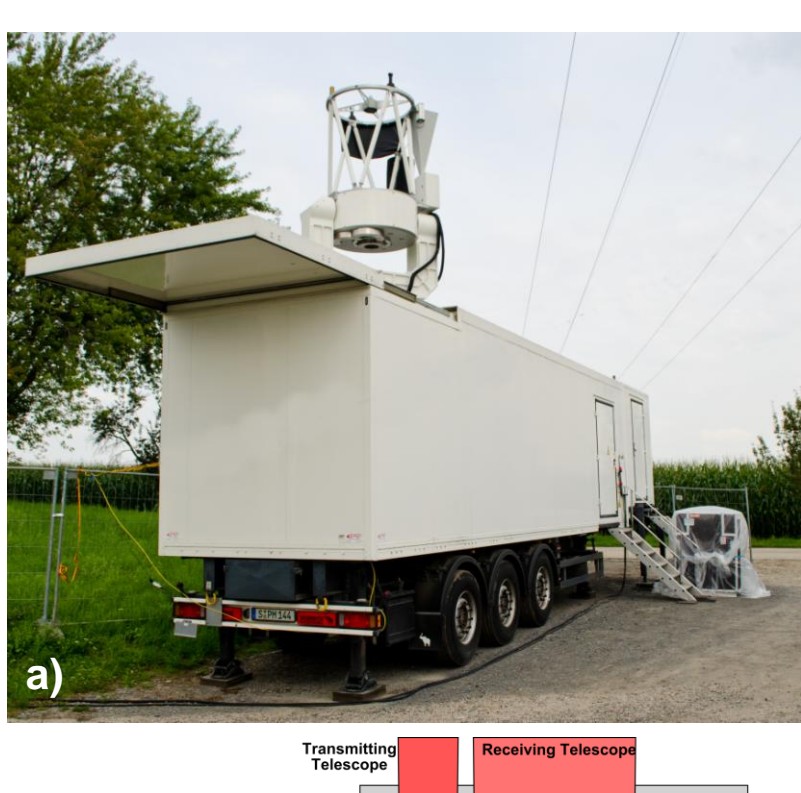

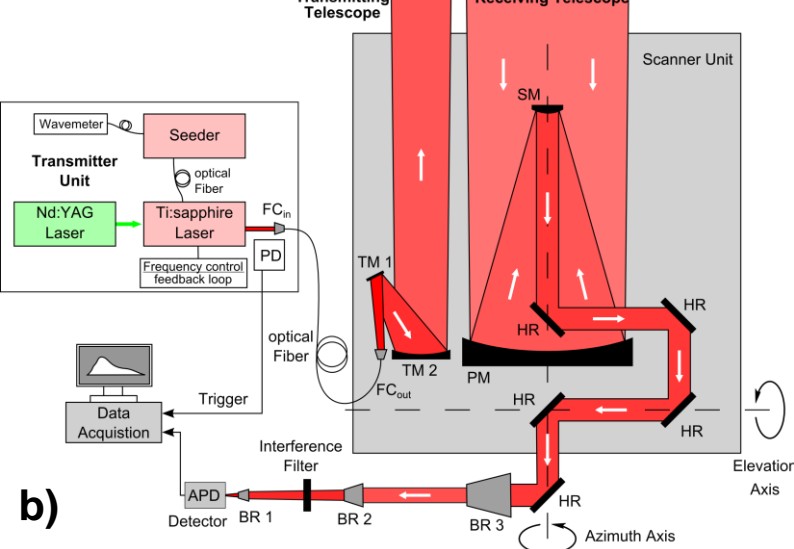

**Figure 1.** a) Photograph of the DIAL system in the field during the SABLE experiment. b) Setup of the UHOH DIAL system with transmitter unit, the scanner unit with transmitting and receiving telescope and the detection path with data acquisition. TM - Transmitting telescope Mirror, PM - Primary Mirror, SM - Secondary Mirror, HR - High-Reflection Mirror, BR - Beam Reducer, APD - Avalanche Photo Diode, PD - Photo Diode, FC - Fiber Coupler.





**Figure 2.** Flow chart of data processing.



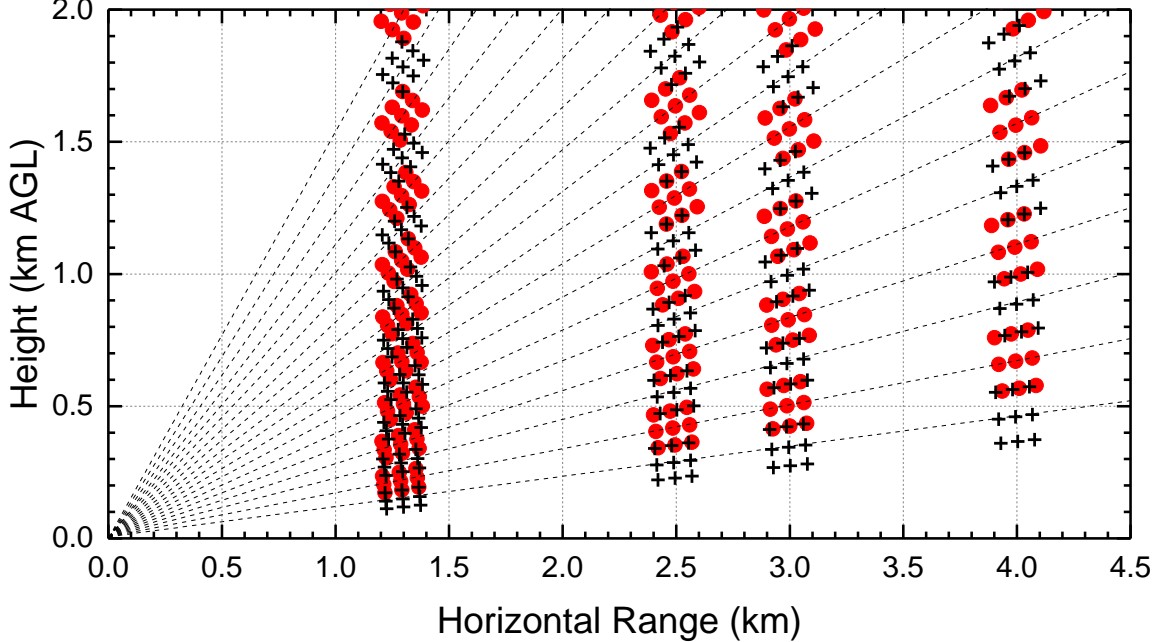

**Figure 3.** Spatial distribution of the grouping of the measured data used for the noise estimation for a single RHI scan. For each group of 9 data points, either black pluses or red dots, the ACF is calculated. Different symbols are used for better understanding when the data points close to each other. Dashed lines indicate selected profiles.





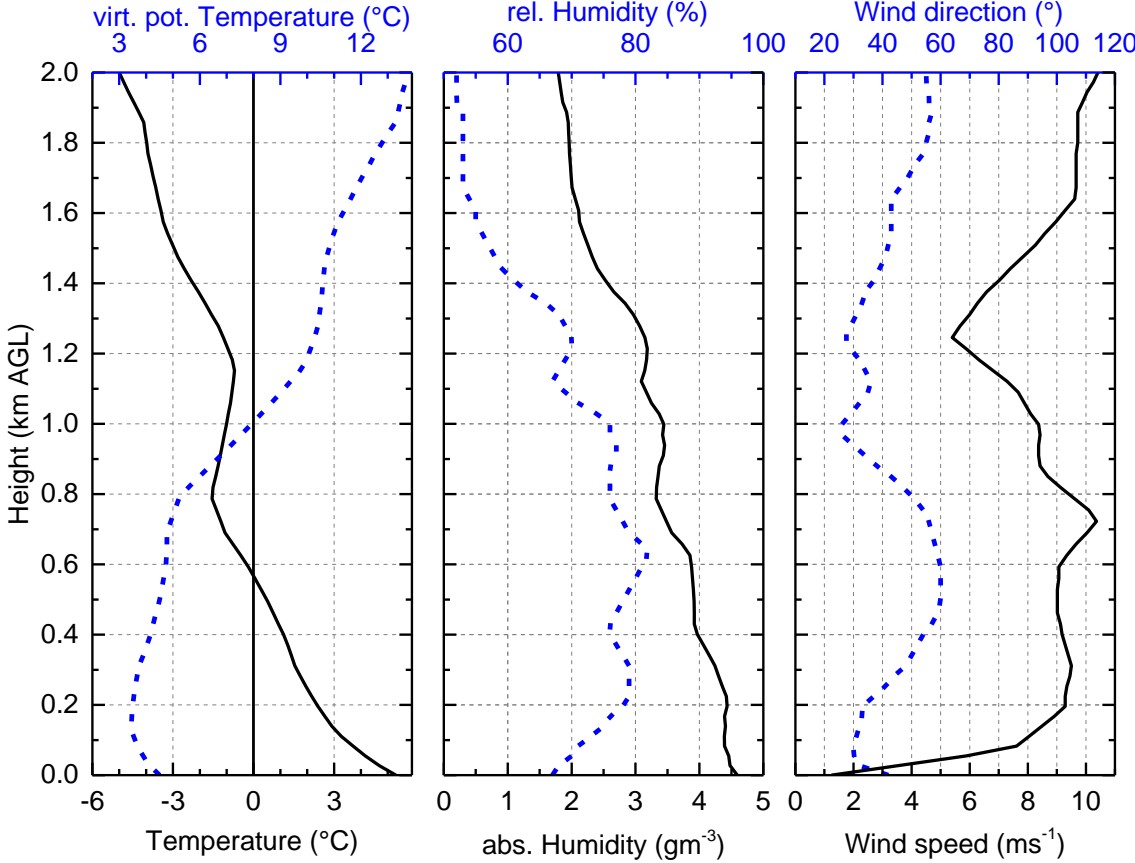

**Figure 4.** Profiles of temperature, virtual potential temperature, absolute and relative humidity, horizontal wind speed and wind direction measured with the radiosonde on IOP 5 on 20 April 2013 at 07:00 UTC.





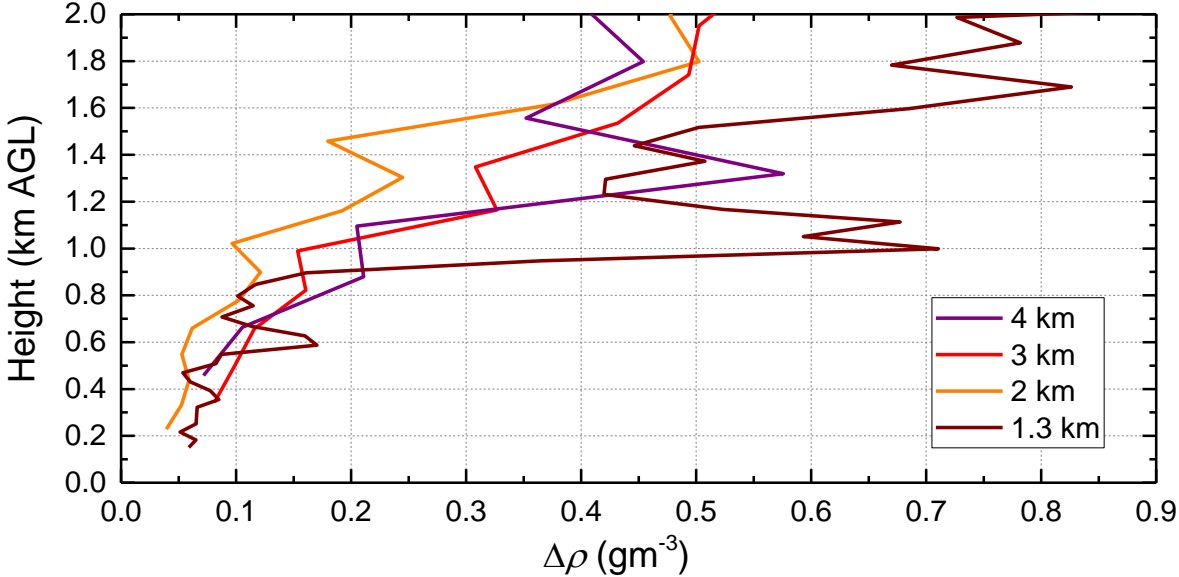

**Figure 5.** Instrumental noise profiles of the RHI scans in Figure 6 for horizontal distances (range resolution) of 1.3 km ($\Delta R$ = 142.5 m), 2 km ($\Delta R$ = 307.5 m), 3 km ($\Delta R$ = 497.5 m), and 4 km ($\Delta R$ = 997.5 m) with a temporal resolution of 10 s.




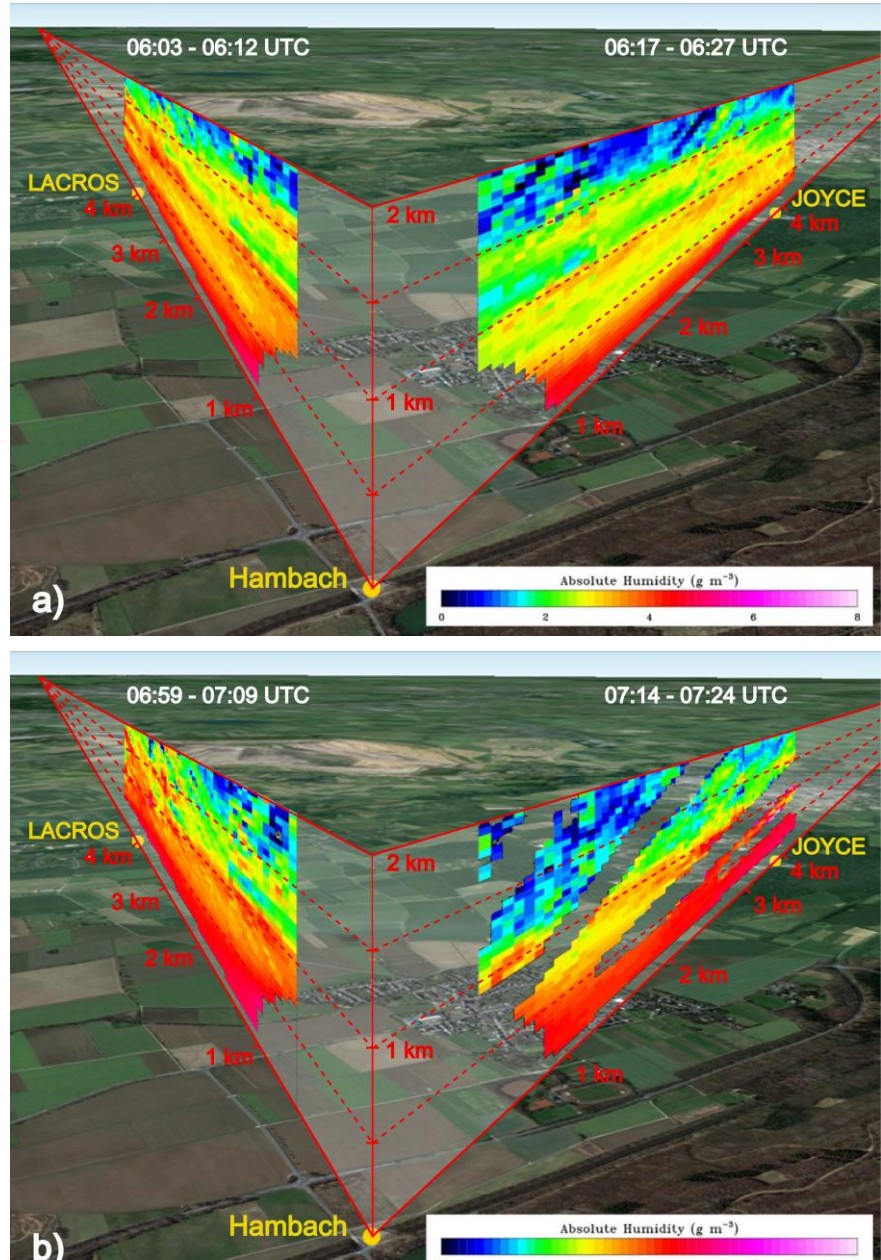

**Figure 6.** 3-dimensional illustration of the WV field in the HOPE domain. The scanning measurements were performed towards LACROS and JOYCE on IOP 5 on 20 April 2013 between a) 06:03 and 06:27 UTC and b) 06:59 and 07:24 UTC. The UHOH DIAL system was located at Hambach site and scanned towards the other supersites LACROS and JOYCE. The distances to the other sites were around 4 km. Background image from Google Earth.



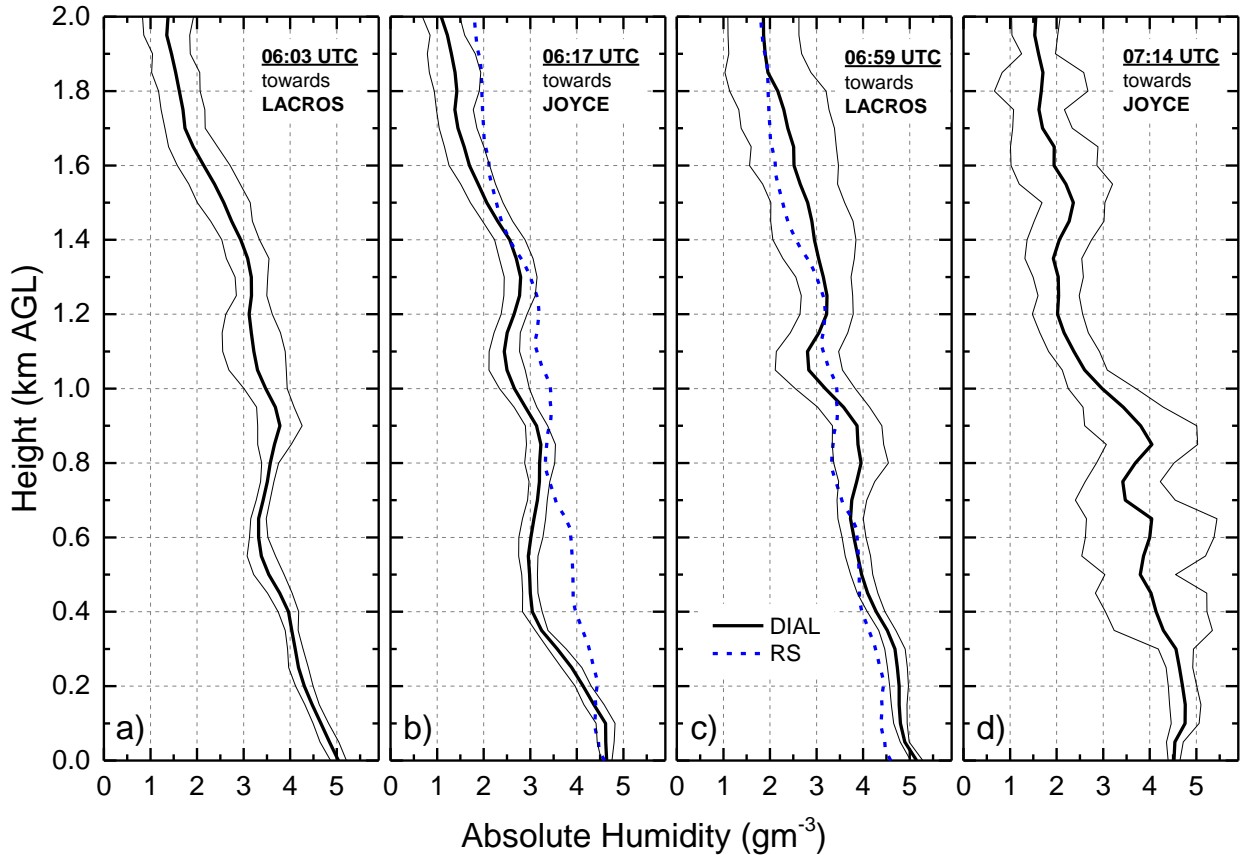

**Figure 7.** Spatially averaged humidity profiles of the scanning measurements of Figure 6. The profiles in a) are of the scans towards LACROS and in b) towards JOYCE. The thick solid lines show the mean profiles of each scan and the thin lines indicate the horizontal variability of humidity ($1\sigma$ standard deviation) within the scan range. The radiosonde profile at 07:00 UTC was plotted with a dashed line with the two closest scans (b and c).





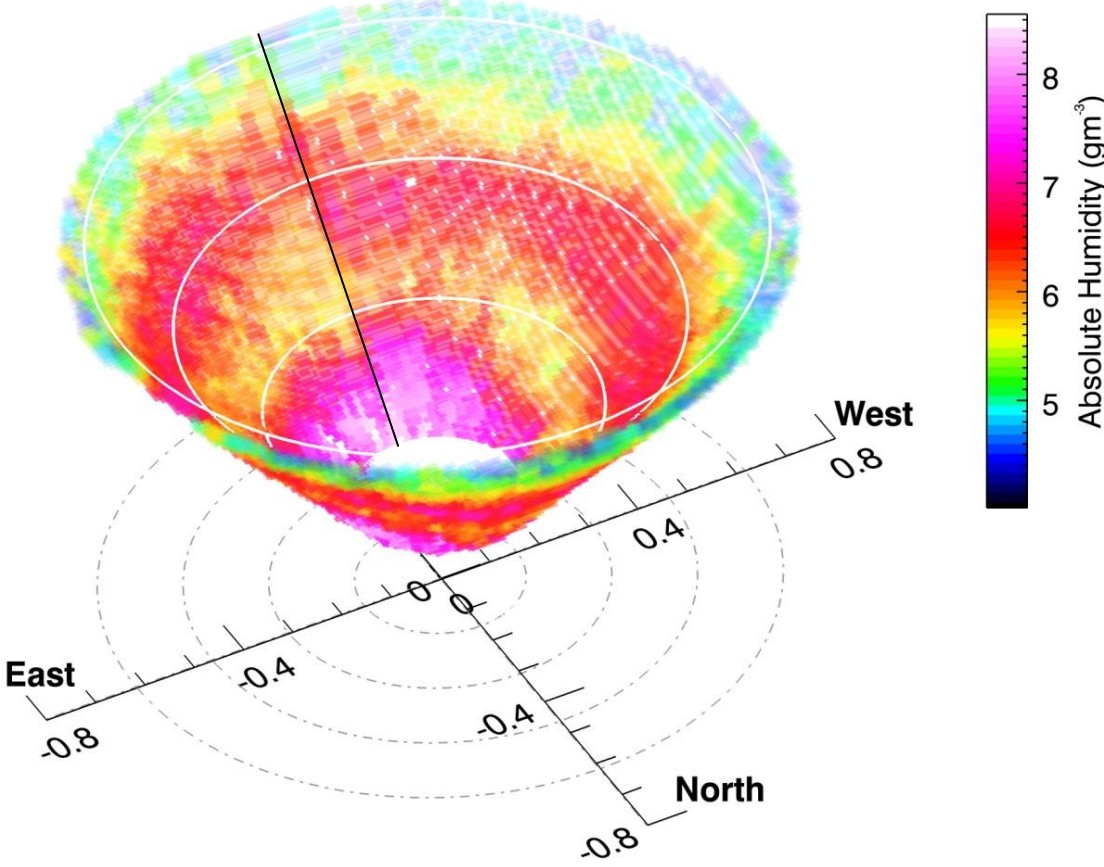

**Figure 8.** Conical scan of SOP 2 on 22 August 2014 between 08:40 and 08:55 UTC. The cone has an elevation angle of 50°. The data were plotted for a height range from 0.2 km up to 0.8 km. The white circles indicate the height in 0.2 km steps and the grey dashed-pointed circles show their projection down to the ground. The black solid line marks the start and end direction of the conical scan; the scanner unit moved clockwise. The scales are given in km.





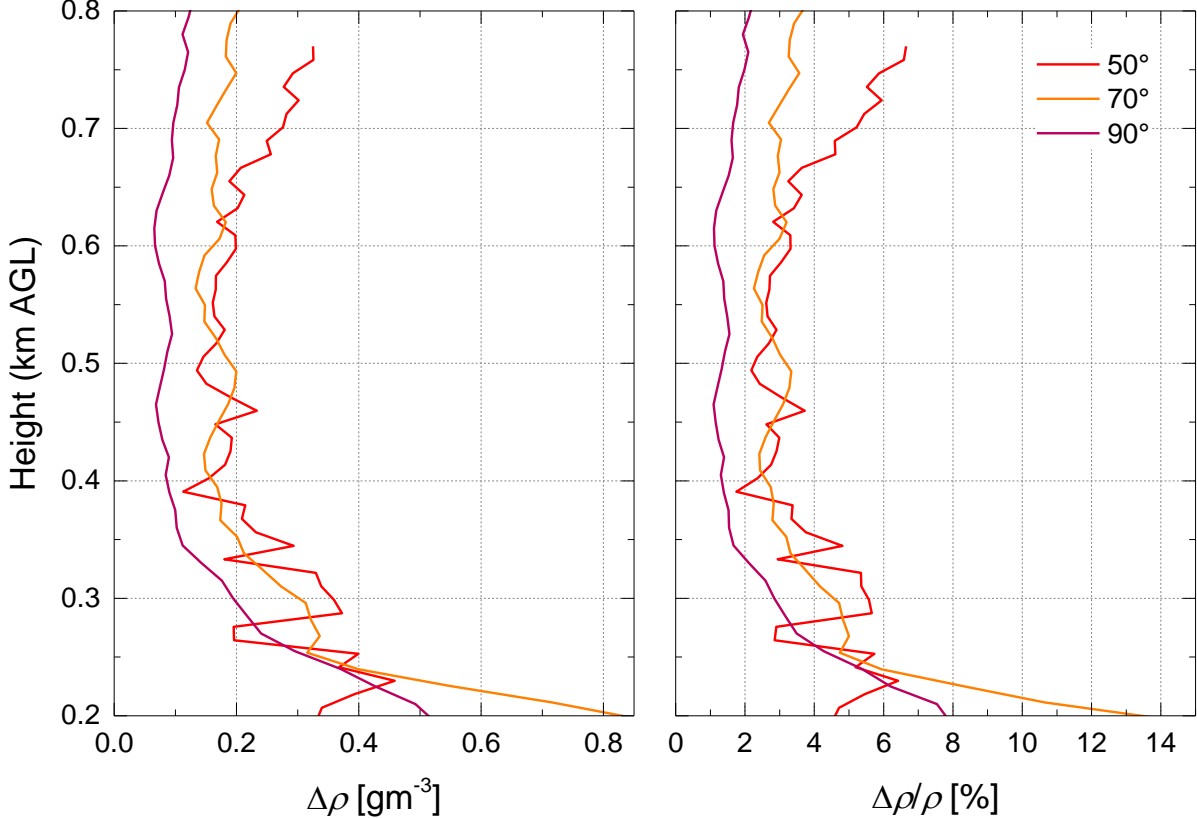

**Figure 9.** Absolute noise $\Delta\rho$ and relative noise $\Delta\rho/\rho$ profiles of selected parts of the volume scan with the corresponding temporal and spatial resolution of 10 s and 67.5 m, respectively.





**Figure 10.** Cutting planes through the 3D data set delivers cross-section images. a) Schematic illustration of the 3D data set with the cutting planes. (b-i) Horizontal and vertical cross-section images of the different cutting plans of a). The black lines in (b-g) indicate the top of the boundary layer. The dotted lines in (d) illustrate the location of the horizontal planes (h-i).





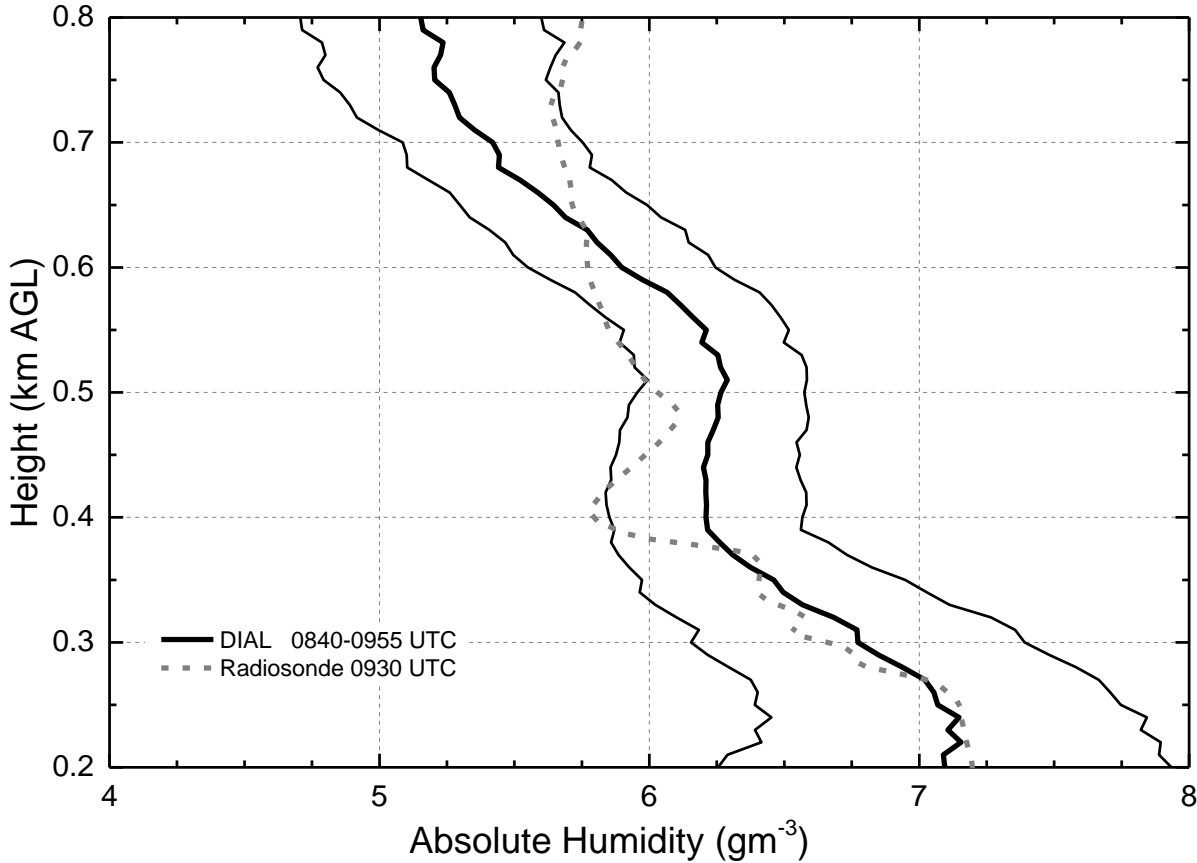

**Figure 11.** Spatially averaged absolute humidity profiles of the volume scan of Figure 10. The thick solid line shows the mean profile of the scanned volume and the thin lines indicate the horizontal WV variability ($1\sigma$ standard deviation) within the scan range. The radiosonde profile at 09:30 UTC was plotted with a dashed line.





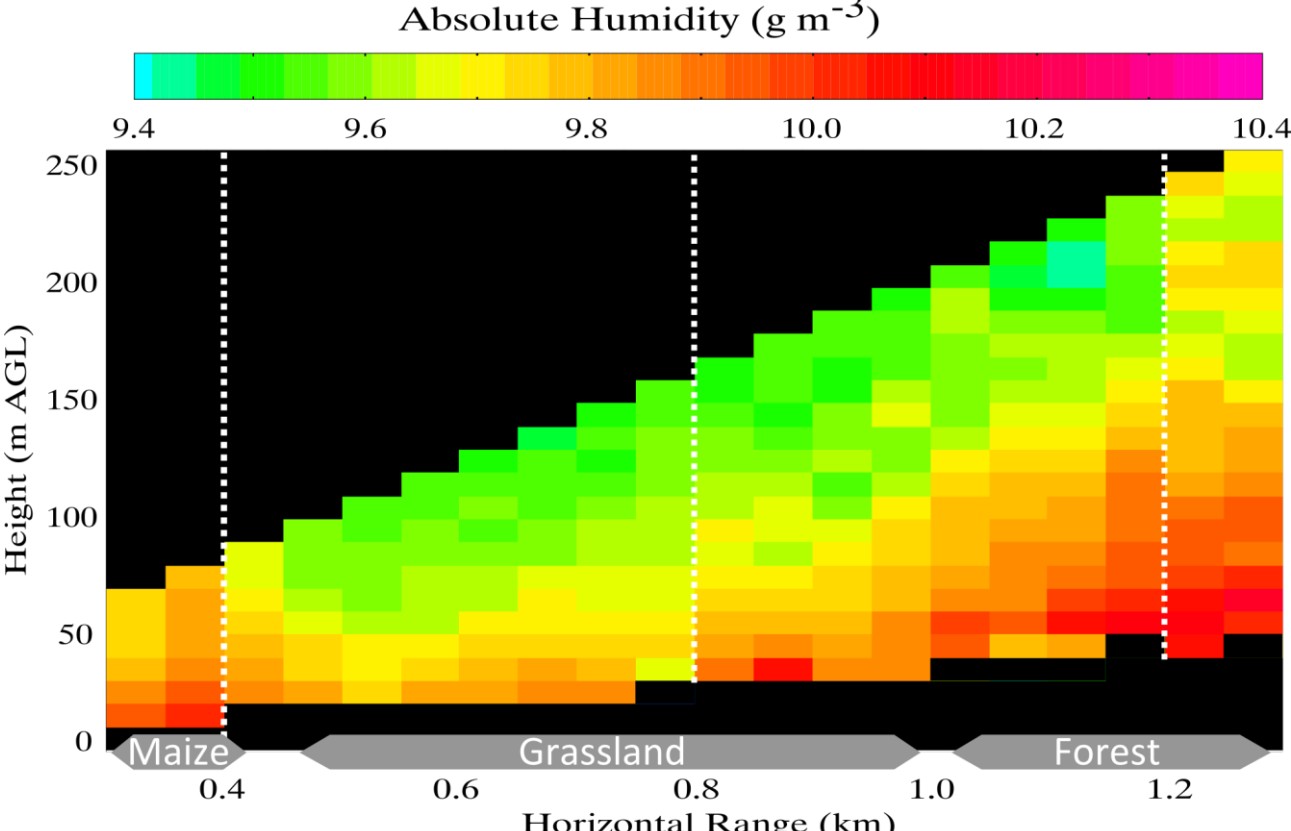

**Figure 12.** One-hour mean WV field between 11:00 and 12:00 UTC on IOP 4 on 12 August 2014. The covered angle range was 0 to 12°, the scan speed was 0.2° s$^{-1}$, a single scan took 1 minute. For the water vapor calculation, 1 s averaged data were used. The land cover along the line-of-sight is also shown. The dotted lines indicate the location of the vertical profiles shown in Figure 13.




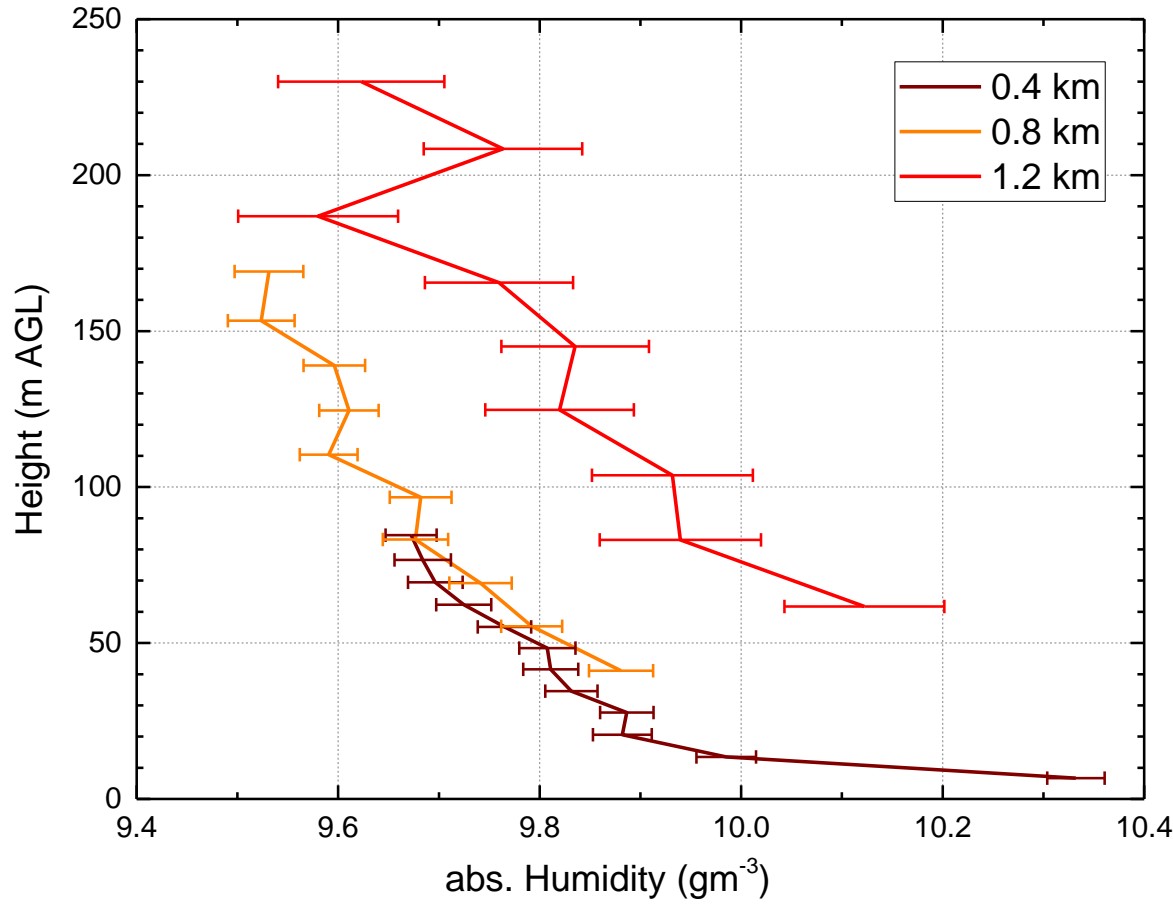

**Figure 13.** Absolute humidity profiles for the RHI scan of Figure 12 for the horizontal distances of 0.4 km, 0.8 km, and 1.2 km. The noise errors are also shown. The location of the WV and noise profiles are indicated by the dotted white lines in Figure 12.