# Peer review of "3D Water Vapor Field in the Atmospheric Boundary Layer Observed with Scanning Differential Absorption Lidar"

_Atmospheric Measurement Techniques, 2015_

## Referee Comment (RC1) · Anonymous Referee #2 · 14 Feb 2016

Please see the review report attached in case the format is not appropriate here:

Review report 3D Water Vapor Field in the Atmospheric Boundary Layer Observed with Scanning Differential Absorption Lidar Author(s): F. Späth et al. MS No.: amt-2015-393 MS Type: Research article

In the manuscript amt-2015-393 authors demonstrated the potential of a new ground-based high-power scanning DIAL system for boundary layer process studies. They also performed detailed error analyses using ACF-based method and compared the results with radiosonde profiles of absolute humidity. The unique aspect of DIAL technique (i.e. no calibration necessary) has been discussed in detail. I found the data analyses tools they have developed for scanning measurements to be unique. Also, the potential of

the visualization tools explored are of huge advantages for studying 2-3 d field of water vapor mixing ratio in the boundary layer. The amount of data set selected from two different campaigns is extensive and sufficient. The results and discussion presented in the manuscript are convincing. However, I have few specific comments on different aspects of the results presented. Since the authors should explore some detailed explanations at various places, major revisions are triggered. After addressing those, the paper deserves publication in the journal AMT. I encourage the authors to consider the comments during revisions. For instance, they used some general statements in their results section.

Specific comments

In Abstract: At the beginning, an overview statement stating the need for a high-resolution 2-3 d field of water vapor information is required. A clear motivation behind such development could be outlined. A brief statement with specific information on the scanning data analyses tools developed and the unique visualization tools used should be made in the abstract as well. Please mention at the beginning that data obtained from two different field campaigns were used for this study. Otherwise, SABLE comes as an afterthought. General statements like "Volume scans show the water vapor field in three dimensions" needs to be removed from the abstract. Please note that I indicated similar general statements made at various places of the manuscript without specified goals or quantifications. I suggest removing those as well (see minor comments). Could you please specify soil type, moisture regime, land cover, vegetation type where you used the phrase "different land use"? Note that this issue has been discussed in the paper. Introduction: Discussion on the comparison between Raman lidar technique and DIAL is extensive. Please shorten this part since you do not compare the results with Raman lidar retrievals with a different instruments. Some references would suffice here. P4 L4: Low elevation. What's the elevation angle? Please mention the overlap issue here and the distance from where WV information is retrieved. Furthermore, while the present study is campaign-based, this dataset should

be highly valuable in future studies involving LES simulations of these events. There-fore, the paper will be a worth addition to the literature. I strongly recommend adding few highlights within an outlook.

Other sections P5 L20: "Firs time.." This need to be mentioned in the abstract Eq 5: There are some discrepancies among Eqns. 4, 5, and relevant discussion and Fig. 2 where Beta_Par is mentioned. In the primitive DIAL equation, Beta_par is not present. I understand that on page 7 you mentioned "Rayleigh-Doppler correction" effect "was not critical as confirmed not only by comparisons with radiosoundings.." Could you please state after the Schotland approximation (i.e. Eqn. 4 without the second term, right?), that this is the equation which has been used as Beta_par is not present in this equation. Or I missed some information here. In any case, all these discussions need to be harmonized and presented at one section. It is not clear also whether or not, you applied the corrections in the measurements shown in the figures. P9 L9: "With small periods" Be specific/quantify. P9 L21: Figure 3: Please explain whether or not any types of interpolation used since for larger ranges (e.g. 3-4 km) at a fixed elevation angle, separation between the circles and crosses would be larger compared to that at closer range (e.g. 1 km) P9 L25: "Finally..." Could you please explain the impact of different temporal averaging here since two different temporal averaging would result in two different angular separation (let's assume r, theta, phi co-ordinate). P11 L17: Below the top. Please quantify P11 L25: Could you please make a note on some of the gaps observed in the 2d field for scan performed between 0714 and 0724 UTC? Also, make a note on the partial overlap effect here. Figure 6a,b: Could you please enlarge the color bar scale limits and color bar title? See Fig. 8 for an example. P12 L8: Could you please make a note on the impact of wind-driven drift of the radiosonde here? P13 L10: Here and elsewhere, for a discussion, could you please mention the changes in the orography in the region? This is important since at many places, you mention this issue. P16-17: An outlook is missing.

Technical corrections/minor comments P1 L6: Add "3d" before fields P1 L13: "range of

a few kilometers" Please also indicate the height here in addition to the range. P2 L17: rephrase the last part with "nor" Used two times. P2 L 19: 2015 a/b: Please check here and throughout the MS. P6 L3: Replace "initial" by "transmitted", "time resolution" by "sampling resolution" Eq 2: áűź0 by áűź? Eq 4: DB: Doppler broadening? P7 L14: "were studying" Rephrase P10: L1: Section 4 header: Tool long for a header title. Please shorten. P10 L31: UTC. Please define w.r.t. local time. P11 L6: 1.6 km. Looks like 1.4 km! P11 L9: 566 s: Please round it to 10 minutes since you do not mention seconds in the time stamps used in the figures. P11 L3: Cirrus: Please mention as found by off-line signals" Right!? P14 L12: Replace "corresponds ..elevation" by "varied according to underlying orography" L30: Heat fluxes: What time? P 15, Section 6.2: Which campaign, which data sets were used need to be mentioned at the beginning. P15 L30: Please make a note on the differences in the terrain/topography.

Remove the following sentences as has been found either general information or repetitions. Please take care on this issue while revising the paper. P3 L 1: "Lidar with. . ..processes" P8 L22: "Thus, the DIAL. . .. . .. .profile"

---

## Author Comment (AC1) · 24 Feb 2016

We thank the reviewer for reading the manuscript in detail and for her/his very helpful comments to improve the manuscript. We would like to respond briefly to two of the comments already here and will give a detailed response to all comments with the revised version.

Concerning the Rayleigh-Doppler correction and Eq. (4) and (5) and Fig. 2: Equation (4) gives the full DIAL equation including the Rayleigh-Doppler (RD) correction term (2nd term). Figure 2 shows the flow chart of the water vapor calculation with all needed parameters to compute the humidity data. This includes also $\beta_{par}$ and $\beta_{mol}$ for the RD correction. If the RD effect is small we skip the additional effort (and related

uncertainties) of the RD correction and apply only the simplified DIAL equation (Eq. (5)), the so-called Schottland approximation.

On page 7, line 24 we discuss the RD correction for the presented cases here and explain that this effect was not critical for two reasons: no large $\beta_{par}$ gradients within the ABL and an online frequency located at the wing of the absorption line. Consequently, Eq. (5) was used here throughout the manuscript.

We are going to clarify this in the revised manuscript.

Concerning the overlap issue and the gaps in the measurements of Fig. 6:

The gaps occur from clouds at the top of the CBL. Lidars cannot measure beyond optically thick clouds. Thus, only measurement data along the line of sight up to the cloud edge are available. In the near range of the RHI scans we omitted the data up to 950 m because of overlap effects (field-of-view of the transmitting and receiving telescopes). This effect could be reduced significantly during the SABLE campaign to distances of only 200-300 m to the lidar.

---

## Referee Comment (RC2) · Anonymous Referee #1 · 17 Mar 2016

The paper discusses different sampling strategies conducted with a scanning water vapor (WV) DIAL (the University of Hohenheim DIAL) to derived 3D WV fields in the lower troposphere. Data from a recent campaign in Germany are used to illustrate the purpose as well as assess the uncertainties associated with the different strategies (RHIs, volume scans and horizontal scans) and the necessary trade-offs between scan speed, temporal and range resolution needed observed the small scale WV structures in the atmosphere. The selected case studies from the SABLE and HOPE campaigns nicely and convincingly illustrate the capability of the Hohenheim WV DIAL.

The paper is well written, well-structured and pleasant to read. My recommendation is that the be published provided that the authors clarify some points listed below (minor

[Figure]

revision).

Abstract P1, lines 11-12: I do not think the sentence "HOPE was part of the project High Definition of Clouds and Precipitation for advancing Climate Prediction (HD(CP)2)" is useful in the abstract.

Introduction

P3, line 16: resulting IN P3, line 17: "Operating the WVRL in the UV but at larger wavelength [...]" not clear. Do you mean at 355 nm? P3, lines 17-19: to be complete on the subject, you should mention that daytime measurements are possible provided that narrow interferential filters are used which are effective to screen out the solar background. The Raman lidar at the ARM site in Oklahoma uses such a technique. P3, lines 24-25: I agree this is a first when using a unique instrument. However, others have combined measurements from different instruments to achieve the 3D documentation of WV in the troposphere.

Section 3 Section 3.2 P8, line 8: how is the background correction made? On a shot-by-shot basis? How do you measure the background light that needs to be subtracted from you backscatter signal? P8, line 11: a senstence starting with "as well as" is lame. Please rephrase. P8, line 21: "must apply" or "must be applied", not "must applied"

Section 3.3 P8, line 25 "WV associated with scanning" or "from scanning"

Section 4 Section 4.1 P10, line 24: "This dataset" rather than "These results" P10, line 31: m s-1 instead of ms-1 P10, line 31: with a minimum of 5.5 m s-1 at 1250 m above ground level (agl). In the following, please use m agl when referring to heights above the measurement sites or above ground. P11, line 5: there is a complex layering in the humidity observations. However, the virtual potential temperature profile shows that the residual layer top is around 800 m, while the developing mixed layer top is at 200 m. There is another temperature inversion around 1600 m that may be related to more synoptic meteorological conditions such as large scale subsidence,etc... P11, line 12:

according to the following procedure differently for different ranges. P11, lines 16-17: "For all distances, the error profiles show a clear increase at the top of the ABL which rises from an altitude of around 0.9 km at 1.3 km distance to 1.1 km at 4km distance." I do not understand this sentence, please clarify. P12, lines 13-14: where are the clouds shown? On Fig. 6? Can you point to or specify in the text where the clouds are?

Section 5 Section 5.1 P13, line 12: use 'm agl' to make sentence less complicated.

Section 5.2 P13, line 22: above 400 km (m not km) P13, line 23: ". . .the noise increases stronger with. . ." this is not clear, please rephrase. P13, line 30: insight INto

Section 6 Section 6.1 P15, line 3: as close as possible TO the land surface P15, line 28: do you mean 800 m?

References Please check carefully: some references are listed which are not cited in the text.

---

## Author Comment (AC2) · 23 Mar 2016

We thank the reviewer for his/her efforts and for the very valuable suggestions and comments. Our detailed responses to all reviewer comments will be submitted in the next days.
* * *